# MULTI-TIMESTEP MODELS FOR MODEL-BASED REINFORCEMENT LEARNING

## ABSTRACT

In model-based reinforcement learning (MBRL), most algorithms rely on simulating trajectories from one-step dynamics models learned on data. A critical challenge of this approach is the compounding of one-step prediction errors as length of the trajectory grows. In this paper we tackle this issue by using a multi-timestep objective to train one-step models. Our objective is a weighted sum of a loss function (e.g., negative log-likelihood) at various future horizons. We explore and test a range of weights profiles. We find that exponentially decaying weights lead to models that significantly improve the long-horizon R2 score. This improvement is particularly noticeable when the models were evaluated on noisy data. Finally, using a soft actor-critic (SAC) agent in pure batch reinforcement learning (RL) and iterated batch RL scenarios, we found that our multi-timestep models outperform or match standard one-step models. This was especially evident in a noisy variant of the considered environment, highlighting the potential of our approach in real-world applications.

## 1 INTRODUCTION

Reinforcement learning is a paradigm where a control agent (or policy) learns through interacting with a dynamic system (or environment) and receives feedback in the form of rewards. This approach has proven successful in addressing some of the most challenging problems globally, as evidenced by Silver et al. (2017b; 2018); Mnih et al. (2015); Vinyals et al. (2019). However, reinforcement learning remains largely confined to simulated environments and does not extend to real-world engineering systems. This limitation is primarily due to two factors: i) the scarcity of data resulting from operational constraints, and ii) the safety standards associated with these systems. Model-based reinforcement learning (MBRL) is an approach that can potentially narrow the gap between RL and applications.

Model-based reinforcement learning (MBRL) algorithms alternate two steps: i) model learning, a supervised learning problem to learn the dynamics of the environment, and ii) policy optimization, where a policy and/or a value function is learned by sampling from the learned dynamics. MBRL is recognized for its sample efficiency, as the learning of policy/value is conducted (either wholly or partially) from imaginary model rollouts (also referred to as background planning), which are more cost-effective and readily available than rollouts in the true dynamics (Janner et al., 2019). Moreover, a predictive model that performs well out-of-distribution facilitates easy transfer of the true model to new tasks or areas not included in the model training dataset (Yu et al., 2020).

While model-based reinforcement learning (MBRL) algorithms have achieved significant success, they are prone to *compounding errors* when planning over extended horizons (Lambert et al., 2022). This issue arises due to the propagation of model bias, leading to highly inaccurate predictions at longer horizons. This can be problematic in real-world applications, as it can result in out-of-distribution states that may violate the physical or safety constraints of the environment. The root of compounding errors is in the nature of the models used in MBRL. Typically, these are one-step models that predict the next state based on the current state and executed action. Long rollouts are then generated by iteratively applying these models, leading to compunding errors. To address this issue, we propose aligning the training objective of these models to optimize the long-horizon error.

Our key contributions are the following:

- We propose a novel training objective for one-step predictive models. It consists in a weighted sum of the loss function at multiple future horizons.
- We perform an ablation study on different strategies to weigh the loss function at multiple future horizons, finding that exponentially decaying weights lead to the best results.
- Finally, we evaluate the multi-timestep models in pure batch and iterated batch MBRL.

## 2 RELATED WORK

**Model-based reinforcement learning**  MBRL has been effectively used in iterated batch RL by alternating between model learning and planning (Deisenroth & Rasmussen, 2011; Hafner et al., 2021; Gal et al., 2016; Levine & Koltun, 2013; Chua et al., 2018; Janner et al., 2019; Kégl et al., 2021), and in the offline (pure batch) RL where we do one step of model learning followed by policy learning (Yu et al., 2020; Kidambi et al., 2020; Lee et al., 2021; Argenson & Dulac-Arnold, 2021; Zhan et al., 2021; Yu et al., 2021; Liu et al., 2021; Benechehab et al., 2023). Planning is used either at decision time via model-predictive control (MPC) (Draeger et al., 1995; Chua et al., 2018; Hafner et al., 2019; Pinneri et al., 2020; Kégl et al., 2021), or in the background where a model-free agent is learned on imagined model rollouts (Dyna; Janner et al. (2019); Sutton (1991); Sutton et al. (1992); Ha & Schmidhuber (2018)), or both. For example, model-based policy optimization (MBPO) (Janner et al., 2019) trains an ensemble of feed-forward models and generates imaginary rollouts to train a soft actor-critic agent.

**Multi-step predictions**  Multi-timestep dynamics modeling was referred to in early works about temporal abstraction (Sutton et al., 1999; Precup et al., 1998) and mixture of timescales models in tabular MDPs (Precup & Sutton, 1997; Singh, 1992; Sutton & Pinette, 1985; Sutton, 1995). It is also partially addressed in the recent MBRL literature by means of recurrent state space models (Hochreiter & Schmidhuber, 1997; Chung et al., 2014; Ha & Schmidhuber, 2018; Hafner et al., 2019; 2021; Silver et al., 2017a). However, these methods rely on heavy computation and approximated inference schemes such as variational inference. The problem of multi-step forecasting was also addressed in the time series literature (Ben Taieb & Bontempi, 2012; Ben Taieb et al., 2012; Venkatraman et al., 2015; Chandra et al., 2021), but none directly aligns the single-step forecasting model training objective to match the multi-step prediction objective, as we propose.

**Fixed/variable-horizon architectures**  Regarding fixed-horizon models, Asadi et al. (2018; 2019) designed a model that has a distinct neural block for each prediction step, which reduces error accumulation but enlarges the model. Whitney & Fergus (2019) demonstrated that a simple fixed-horizon model has better accuracy and can be used for planning with CEM. However, they had to change the reward function since they did not have access to all the visited states when planning with a fixed-horizon model (discussed in section 4.2). For multi-horizon architectures, Lambert et al. (2021) proposed a model that takes the prediction horizon and the parameters of the current policy as input, instead of the full action sequence. Meanwhile, Whitney & Fergus (2019) used a dynamic-horizon architecture by padding the shorter action sequences with zeros to fit the input size constraint.

**Compounding errors in MBRL**  Lambert et al. (2022) showed that the stability of the physical system affects the error profile of a model. Talvitie (2014) proposed *hallucinated replay*, which inputs to the model its own generated (noisy) outputs. This technique teaches the model to recover from its own errors (similar to denoising), and produces models that have lower one-step accuracy but higher returns in some scenarios. A similar idea (multi-step error) is used by Byravan et al. (2021) to train models in the context of MPC, and by Venkatraman et al. (2015) who casts the error-recovery task as imitation learning. In the same vein, Xiao et al. (2023) learned a criterion based on the $h$-step error of the model, and adjusted the planning horizon $h$ accordingly.

## 3 PRELIMINARIES

The conventional framework used in RL is the finite-horizon Markov decision process (MDP), denoted as $\mathcal{M} = \langle \mathcal{S}, \mathcal{A}, p, r, \rho_0, \gamma \rangle$. In this notation, $\mathcal{S}$ is the state space and $\mathcal{A}$ is the action space.

The transition dynamics, which could be stochastic, are represented by $p : \mathcal{S} \times \mathcal{A} \rightsquigarrow \mathcal{S}$. The reward function is denoted by $r : \mathcal{S} \times \mathcal{A} \to \mathbb{R}$. The initial state distribution is given by $\rho_0$, and the discount factor is represented by $\gamma$, and lies within the range of [0,1].

The objective of RL is to identify a policy $\pi(s)$ which is a distribution over the action space ($\mathcal{A}$) for each state ($s \in \mathcal{S}$). This policy aims at maximizing the expected sum of discounted rewards, denoted as $J(\pi, \mathcal{M}) := \mathbb{E}_{s_0 \sim \rho_0, a_t \sim \pi, s_{t>0} \sim p}[\sum_{t=0}^{H} \gamma^t r(s_t, a_t)]$, where $H$ represents the horizon of the MDP. Under a given policy $\pi$, the state-action value function (also known as the Q-function) at a specific $(s, a) \in \mathcal{S} \times \mathcal{A}$ pair is defined as the expected sum of discounted rewards. This expectation starts from state $s$, takes action $a$, and follows the policy $\pi$ until termination: $Q^\pi(s, a) = \mathbb{E}_{a_{t>0} \sim \pi, s_{t>0} \sim p}\left[\sum_{t=0}^{H} \gamma^t r(s_t, a_t) \mid s_0 = s, a_0 = a\right]$. Similarly, we can define the state value function by taking the expectation with respect to the initial action $a_0$: $V^\pi(s) = \mathbb{E}_{a_t \sim \pi, s_{t>0} \sim p}\left[\sum_{t=0}^{H} \gamma^t r(s_t, a_t) \mid s_0 = s\right]$.

**Model-based RL (MBRL)** algorithms address the supervised learning challenge of estimating the dynamics of the environment $\hat{p}$ (and sometimes also the reward function $\hat{r}$) from data collected when interacting with the real system. The loss function is typically the log-likelihood $\mathcal{L}(\mathcal{D}; \hat{p}) = \frac{1}{N} \sum_{i=1}^{N} \log \hat{p}(s_{t+1}^i | s_t^i, a_t^i)$ or Mean Squared Error (MSE) for deterministic models. The learned model can subsequently be employed for policy search under the MDP $\widehat{\mathcal{M}} = \langle \mathcal{S}, \mathcal{A}, \hat{p}, r, \rho_0, \gamma \rangle$. This MDP shares the state and action spaces $\mathcal{S}, \mathcal{A}$, reward function $r$, with the true environment $\mathcal{M}$, but learns the transition probability $\hat{p}$ from the dataset $\mathcal{D}$. The policy $\hat{\pi} = \arg\max_\pi J(\pi, \widehat{\mathcal{M}})$ learned on $\widehat{\mathcal{M}}$ is not guaranteed to be optimal under the true MDP $\mathcal{M}$ due to distribution shift and model bias.

In pure batch (or offline) RL, we are given a set of $N$ transitions $\mathcal{D} = \{(s_t^i, a_t^i, r_t^i, s_{t+1}^i)\}_{i=1}^{N}$. These transitions are generated by an unknown behavioral policy, $\pi^\beta$. The challenge in pure batch RL is to learn a good policy in a single shot, without further interacting with the environment $\mathcal{M}$, even though our objective is to optimize $J(\pi, \mathcal{M})$ with $\pi \neq \pi^\beta$.

Similarly to Chua et al. (2018); Kégl et al. (2021), we learn a probabilistic dynamics model with a prediction and an uncertainty on the prediction. Formally, $s_{t+1} \rightsquigarrow \hat{p}_\theta(s_t, a_t) = \mathcal{N}\big(\mu_\theta(s_t, a_t), \sigma_\theta(s_t, a_t)\big)$, where $\mathcal{N}$ is a multivariate Gaussian, and $\theta$ is the learned parameters of the predictive model. In practical applications, fully connected neural networks are often employed due to their proven capabilities as powerful function approximators (Nagabandi et al., 2018; Chua et al., 2018; Kégl et al., 2021), and their suitability for high-dimensional environments over simpler nonparametric models such as Gaussian processes. Following previous research (Chua et al., 2018; Kégl et al., 2021), we assume a diagonal covariance matrix for which we learn the logarithm of the diagonal entries $\sigma_\theta = \text{Diag}(\exp(l_\theta))$, where $l_\theta$ denotes the output of the neural network.

## 4 METHODS

### 4.1 PROBLEM STATEMENT

In MBRL it is common to use a model $\hat{p}$ that predicts the state one-step ahead $s_{t+1} \rightsquigarrow \hat{p}(s_t, a_t)$. We train this model to optimize the one-step predictive error $L\big(s_{t+1}, \hat{p}(s_t, a_t)\big)$ (MSE or NLL for stochastic modeling) in a supervised learning setting. To learn a policy, we use these models for planning $h$ steps ahead by applying a procedure called *rollout*: we generate $s_{t+j} \rightsquigarrow \hat{p}(s_{t+j-1}, a_{t+j-1})$ recursively for $j = 1, \ldots, h$. Here $(a_t, \ldots, a_{t+h-1}) = \boldsymbol{a}_{t:t+h}$ is either a fixed action sequence generated by planning or sampling policy $a_{t+j} \rightsquigarrow \pi(s_{t+j})$ for $j = 1, \ldots, h$, on the fly. Formally, let

$$
\begin{aligned}
\hat{p}^1(s_t, a_t) &= \hat{p}(s_t, a_t) \text{ and} \\
\hat{p}^j(s_t, \boldsymbol{a}_{t:t+j}) &= \hat{p}\big(\hat{p}^{j-1}(s_t, \boldsymbol{a}_{t:t+j-1}), a_{t+j-1}\big) \text{ for } j = 2, \ldots, h.
\end{aligned}
\tag{1}
$$

Using $\hat{p}^h(s_t, \boldsymbol{a}_{t:t+h})$ to estimate $s_{t+h}$ is problematic for two reasons:

- A distribution mismatch occurs between the inputs that the model was trained on (sampled from the true unknown transition distribution) and the inputs the model being evaluated on (sampled from the predictive distribution of the model; Talvitie (2014; 2017)).

- The predictive error (and the modeled uncertainty in the case of stochastic models) will propagate through the successive model calls, leading to compounding errors (Lambert et al., 2022; Talvitie, 2014; Venkatraman et al., 2015).

To mitigate these issues, we propose models that learn to predict the state $s_{t+h}$ after an arbitrary future horizon $h$ given $s_t$ and an action sequence $\boldsymbol{a}_{t:t+h}$. We identify two levels in addressing the stated problem:

1. First we build *direct* models $\hat{p}_h(s_t, \boldsymbol{a}_{t:t+h})$ that predict the state at a fixed future horizon $h$ (similar to Whitney & Fergus (2019)). These models cannot directly be used for planning since they do not give access to intermediate rewards $(r_{t+1}, \ldots, r_{t+h-1})$, but they serve to motivate the second level, our **main contribution**:

2. We build one-step models $\hat{p}$ but train them to predict the $h$-step error (2) using the recursive formula (1). Since $\hat{p}$ is a neural net, here we are essentially using recurrent nets.

We start our analysis from Level 1 in Section 4.2, and address Level 2 in Section 4.3. Finally, the experimental setup and the performance evaluation of the agents and our proposed models is showcased in Section 5.

## 4.2 FIXED-HORIZON MODELS

Fixed-horizon models $\hat{p}_h$ take as input the current state $s_t$ and an action sequence $\boldsymbol{a}_{t:t+h} = (a_t, \ldots, a_{t+h-1})$, and predict the parameters of a diagonal multi-variate Gaussian distribution over the state observed $h$ steps into the future: $\hat{s}_{t+h} \sim \mathcal{N}(\hat{\mu}_\theta(s_t, \boldsymbol{a}_{t:t+h}), \hat{\sigma}_\theta(s_t, \boldsymbol{a}_{t:t+h}))$ with $\theta$ representing the parameters of $\hat{p}_h$.

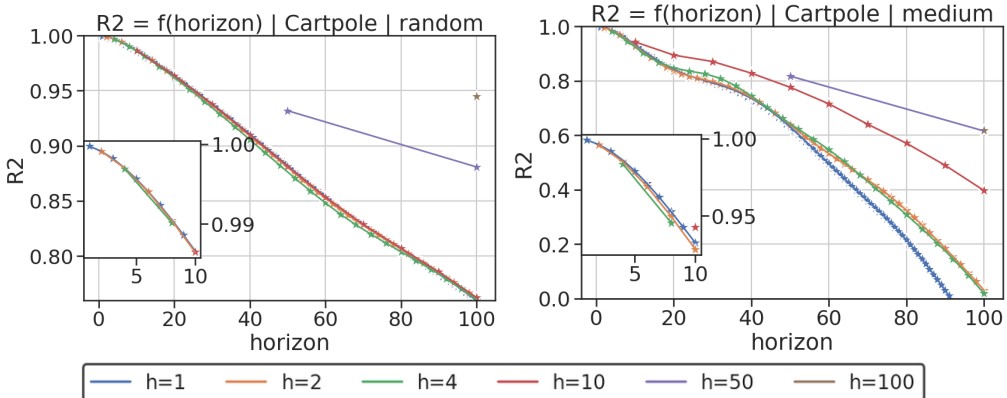

Figure 1: R2 score (explained variance) as a function of the prediction horizon for several multi-step models. We use prediction horizons $h \in \{1, 2, 4, 10, 50, 100\}$, and apply the predictors recursively according to (1) up to maximum horizon $H = 100$. The difference between the two plots is the evaluation dataset: *random* (where the behaviour policy is a random policy) and *medium* (where the behaviour policy is a half-expert policy). A discussion on the differences between these datasets is given in Appendix A.2.

Figure 1 shows the error at different horizons for different fixed-horizon models and datasets. For each horizon $h$, the error is computed by considering all sub-trajectories of size $h$ from the test set. We call each $h$-step model $\hat{p}_h$ recursively $H/h$ times to compute the prediction $\hat{p}_h^{H/h}$ up to $H$ steps ahead.

In general, for a given horizon, we observe that the R2 score (explained variance) increases as we go from the one-step model to the hundred-step model. For the Cartpole random dataset, the short-horizon models are almost identical, while the fifty- and the hundred-step models achieve distinguishable R2 scores. In the medium dataset the improvement grows steadily with $h$.

### 4.3 TRAINING THE ONE-STEP MODELS WITH THE ERROR $h$ STEPS AHEAD

Figure 1 shows that it may be worthy to train models for longer horizons. But, since our goal is to plan with an arbitrary horizon, we wanted to *improve* the one-step predictor $\hat{p}$ rather than throwing it away. An obvious solution was to train the recurrent predictor $\hat{p}^h$ to minimize $L\big(s_{t+h}, \hat{p}^h(s_t, \boldsymbol{a}_{t:t+h})\big)$, but we found that this made the one-step prediction worse, even for the minimal case of $h = 2$ (Figure 2 with $\alpha = 0$). As can be seen from the same figure, in both medium and random datasets the models where $\alpha < 1.0$, suffer a worse predictive error for short horizons, but recover and beat the vanilla one-step model down-the-horizon.

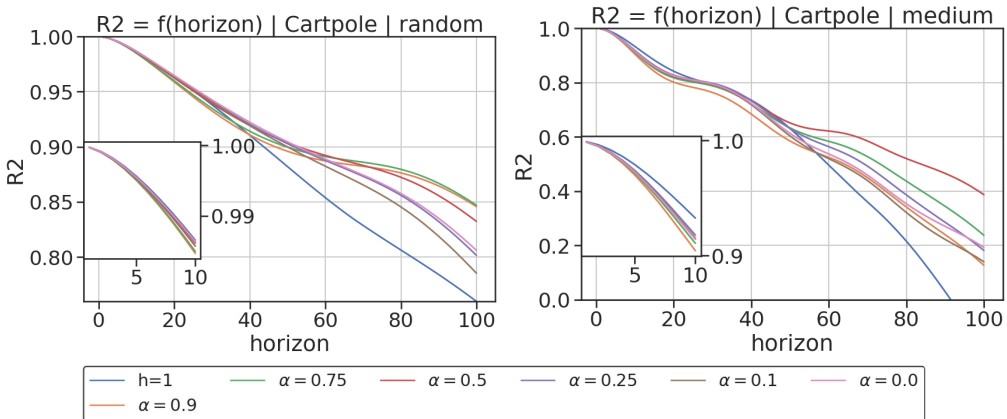

Figure 2: R2 score (explained variance) as a function of the prediction horizon for different weight vectors of $\boldsymbol{\alpha} = [\alpha, 1 - \alpha]$. $\hat{p}$ is learned minimizing $\alpha \times L\big(s_{t+1}, \hat{p}(s_t, a_t)\big) + (1 - \alpha) \times L\big(s_{t+2}, \hat{p}(\hat{p}(s_t, a_t), a_{t+1})\big)$.

Following the empirical findings in the previous simple case, we hypothesize that the solution is to force the predictor $\hat{p}$ to be good on *all* horizons. Formally, we define horizon-dependent weights $\boldsymbol{\alpha} = (\alpha_1, \ldots, \alpha_h)$, and minimize the loss

$$L_{\boldsymbol{\alpha}}\big(\boldsymbol{s}_{t+1:t+h+1}, \hat{p}^{1:h}(s_t, \boldsymbol{a}_{\cdot})\big) = \sum_{j=1}^{h} \alpha_j L\big(s_{t+j}, \hat{p}^j(s_t, \boldsymbol{a}_{t:t+j})\big). \tag{2}$$

To proceed with *gradient descent*-based optimization, we only need the gradient of the loss with respect to the model's parameters. Therefore, we provide an analysis of the analytical gradient of this loss in Appendix C. We derive the gradient in equation 6 to highlight the fact that unlike existing literature (hallucinated replay Talvitie (2014) and multi-step loss Byravan et al. (2021)), our proposed method consists in back-propagating the gradient of the loss through the successive compositions of the model ($\hat{p}^{1:h}(s_t, \boldsymbol{a}_{\cdot})$) figuring in $L_{\boldsymbol{\alpha}}$. Furthermore, we find that the derivative of the generalized loss $L_{\boldsymbol{\alpha}}$ can be expressed as a linear function of the derivative of the loss one-step ahead $L\big(s_{t+1}, \hat{p}(s_t, a_t)\big)$. This opens the door for gradient approximation ideas that are briefly discussed in Appendix C, and intended to be explored in future follow-up works.

An important matter that remains to solve is the choice of the weights $\{\alpha_j\}_{j \in \{1, \ldots, h\}}$. For instance, Figure 2 shows that for $h = 2$ the best weight vector $\boldsymbol{\alpha} = (\alpha, 1 - \alpha)$ depends on both the horizon and the dataset, and that it is neither $[1, 0]$ (the classical $\hat{p}$ trained for one-step error), nor $[0, 1]$ ($\hat{p}$ trained to minimize $L\big(s_{t+2}, \hat{p}(\hat{p}(s_t, a_t), a_{t+1})\big)$). In the next section, we present multiple choices for the weights $\alpha_j$, along with the intuition behind each one of them.

#### 4.3.1 HOW TO CHOOSE $\{\alpha_j\}_{j \in \{1, \ldots, h\}}$ ?

The choice of $\alpha_j$ is very important in this context as it allows us to balance the optimization problem between losses at different prediction horizons. In fact, the different loss terms are at different scales

since the error grows with the prediction horizon, and therefore a sensible choice of these weights is critical.

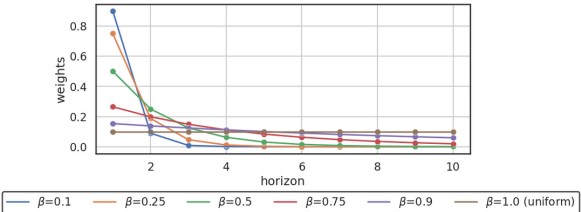

Figure 3: For a maximum horizon of 10, the figure shows the different weight profiles that we get by varying the decay parameter $\beta$.

- **Uniform.** $\alpha_j = 1/h$ The most intuitive way to handle the weighting of the losses is by using uniform weights. However, this method can be sub-optimal since the loss terms $L\big(s_{t+j}, \hat{p}^j(s_t, \boldsymbol{a}_{t:t+j})\big)$ grow with $j$ so in a flat sum long horizon errors will dominate.

- **Decay.** $\alpha_j = (\frac{1-\beta}{1-\beta^{h+1}})\beta^j$ This method implements an exponential decay through a parameter $\beta$ (Figure 3 shows the weight profiles obtained using different values of $\beta$).

- **Learn.** $\alpha_j = learnable$ In this setting, the $\alpha_j$ are left as free-parameters to be learned by the model. The expected outcome in this case is that the model puts the biggest weight on the one-step error $L\big(s_{t+1}, \hat{p}(s_t, a_t)\big)$ as it's smaller than further horizons.

- **Proportional.** $\alpha_j \sim \frac{1}{L\big(s_{t+j}, \hat{p}^j(s_t, \boldsymbol{a}_{t:t+j})\big)}$ This method normalizes all the loss terms with their amplitude so that they become equivalent optimization-wise.

## 5 EXPERIMENTS & RESULTS

This section starts by presenting the experimental setup, followed by the experiments we conducted to evaluate the proposed models.

### 5.1 EXPERIMENTAL SETUP

We are interested in testing our models in two ways: "static" model learning from a fixed data set, and "dynamic" agent learning from model-generated traces. While we are interested in building models that improve the predictive error beyond the one-step transition, the relationship between this surrogate metric and the return of the underlying agent is not trivial. Indeed, in many scenarios, models that have lower predictive ability lead to better performing agents (Talvitie, 2014).

Therefore, we suggest the following experimental settings:

- **Pure batch (offline) RL.** In this setting, we dispose of multiple datasets with different characteristics. We then test the predictive error of our models against these datasets in a supervised-learning fashion. The main metric in this setting is the R2 score that we evaluate at different future horizons. The resulting models can then be used to train an RL agent (We use Soft-Actor Critic, implementation details can be found in Appendix B) in what can be seen as a single-iteration MBRL.

- **Iterated batch RL.** In this setting, we alternate between generating a system trace (one episode from the real system), training the model using the traces collected so-far, and training the policy (RL agent) on the freshly updated model. We measure dynamic metrics such as the convergence pace and the mean asymptotic reward or return.

We conduct our experiments in the continuous control environment *Cartpole swing-up*. We use the implementation of Deepmind Control (Tassa et al., 2018) which is based on the Mujoco physics simulator (Todorov et al., 2012). A description of this environment can be found in Appendix A. A detailed comparison of the static datasets used in the pure batch RL setting is given in Appendix A.2.

**Noisy Environment** In an effort to further examine the applicability of multi-timestep models, we suggest the use of a *Cartpole swing-up* variant that is characterized by the addition of Gaussian noise with a fixed standard deviation (equal to $1\%$ of the range of a given variable) to all observable variables. The Noisy Cartpole environment is of particular interest due to the potential for multi-timestep models to experience significant failures as a result of the accumulation of substantial noise on the learning targets projected $h$ steps ahead. We aim at demonstrating that our proposed method effectively addresses this problem by introducing self-consistency through the concurrent optimization of errors for future horizons.

## 5.2 STATIC EVALUATION: R2 ON FIXED DATASETS

For each horizon $j$, the error is computed by considering all the sub-trajectories of size $j$ from the Test dataset. The predictions are then computed by $j$ model compositions (using the groundtruth actions) and the average R2 score is reported.

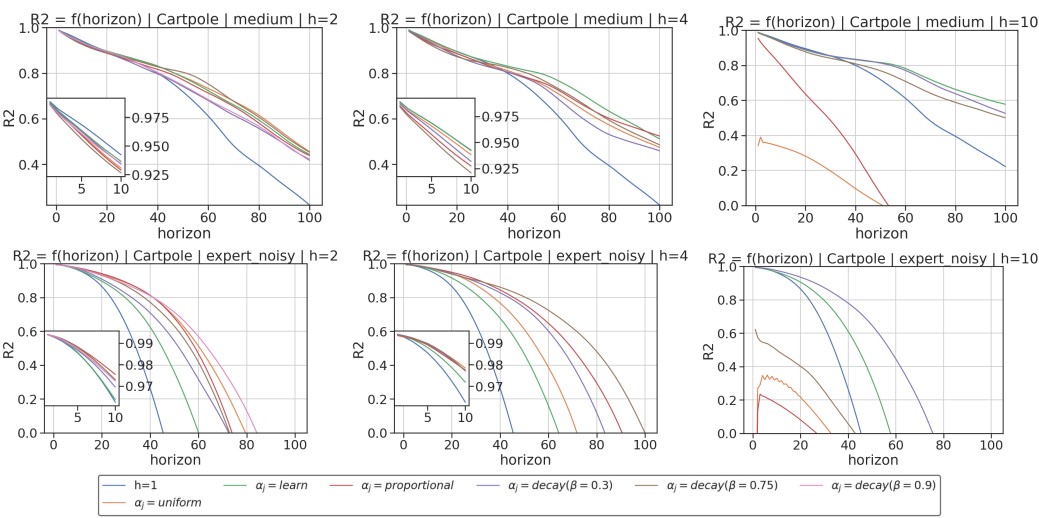

Figure 4: For different maximum horizons $h \in \{2, 4, 10\}$ represented by columns, and datasets (medium and expert_noisy) represented by rows, we show the R2 score as a function of the prediction horizon for different weight profiles.

Figure 4 shows the R2 score as a function of the prediction horizon, when using multiple weight profiles for three maximum horizons models: $h \in \{2, 4, 10\}$. In both the medium and expert_noisy datasets, the multi-timestep models largely improve over the vanilla one-step model. Depending on the future horizons scope, we obtain different optimal weight profiles (mostly from the noisy dataset as in the noiseless one the differences are minor): $decay(\beta = 0.9)$ for $h = 2$, $decay(\beta = 0.75)$ for $h = 4$, and $decay(\beta = 0.3)$ for $h = 10$. We interpret the effectiveness of the *decay* weighting schema from the error profile witnessed for the one-step model. Indeed, the error for this model grows exponentially with the prediction horizon (a theoretical justification can be found in Theorem 1 of Venkatraman et al. (2015)), suggesting that exponentially decaying weights (modulo the right parameter $\beta$) can correctly balance the training objective.

Inspired by this finding, we investigated the direct use of weights that are the exact inverse of the loss term they correspond to ($\alpha_j = proportional$). However, this technique didn't further improve the results. A potential explanation is the interdependence between the different losses as reducing one loss can indirectly affect another one.

Unsurprisingly, the learnable weighting profile quickly converges to putting most of the weight on the one-step error, this is expected because the one-step loss is the smallest among the other terms. A potential fix to this phenomena is to add a regularization term that enforces the weights to be more equally distributed. The elaboration of this idea is beyond the scope of this work.

## 5.3 DYNAMIC EVALUATION: PURE BATCH (OFFLINE) RL

To demonstrate the effectiveness of multi-timestep models in practical RL scenarios, we start with the pure batch RL setting. Using the medium, random, and **expert** datasets for the *Cartpole swing-up* environment, and the expert_noisy dataset for its noisy variant. We train SAC agents on models that are themselves trained on these respective datasets, and evaluate them on the real system. Regarding the multi-timestep models, we choose the best model on each maximum horizon ($decay(\beta = 0.9)$ for $h = 2$, $decay(\beta = 0.75)$ for $h = 4$, and $decay(\beta = 0.3)$ for $h = 10$), and the vanilla one-step model for comparison.

Table 1: Pure batch (offline) RL evaluation: mean $\pm 90\%$ Gaussian confidence interval over 3 seeds. The reported score is the episodic return after training SAC on the pre-trained models for $500k$ steps

| Model | medium | expert_noisy | random | expert |
|---|---|---|---|---|
| $h = 1$ | $813.4 \pm 19.8$ | $580.3 \pm 56.7$ | $308.7 \pm 20.4$ | $807.3 \pm 57.1$ |
| $h = 2 \mid \alpha = decay(\beta = 0.9)$ | $812.8 \pm 32.2$ | $538.8 \pm 75.8$ | $214.5 \pm 71.1$ | $830.6 \pm 44.2$ |
| $h = 4 \mid \alpha = decay(\beta = 0.75)$ | $832.0 \pm 35.7$ | $639.09 \pm 22.5$ | $178.9 \pm 29.2$ | $822.1 \pm 56.6$ |
| $h = 10 \mid \alpha = decay(\beta = 0.3)$ | $842.4 \pm 36.2$ | $557.7 \pm 141.2$ | $208.3 \pm 42.4$ | $730.2 \pm 147.8$ |

Although the results are not yet statistically significant due to a limited number of seeds, multi-timestep models improve over the one-step model in the datasets that require long-horizon accuracy (all but the random dataset where the policy keeps navigating the same region around the initial states).

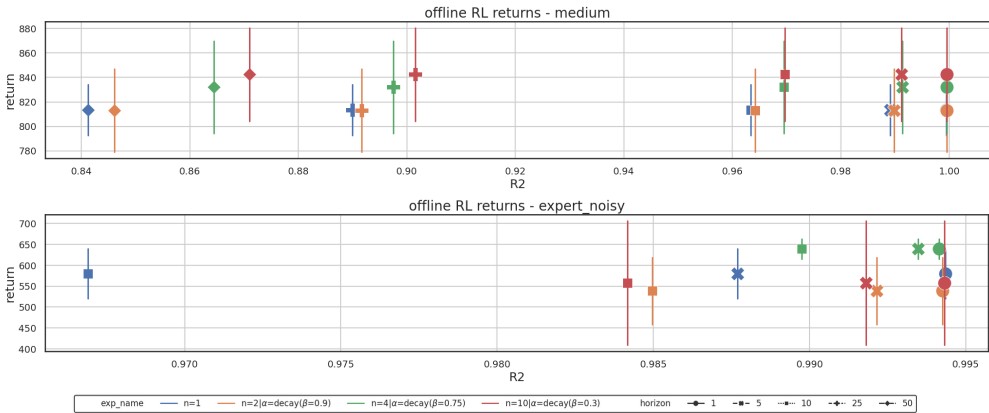

Figure 5: The average episodic return as a function of the R2 score at different horizons $h \in \{1, 5, 10, 25, 50\}$. We show this for both the medium dataset representing the vanilla Cartpole environment, and expert_noisy for its noisy variant. We omit the longer horizons (25 and 50) in the second plot as the corresponding R2 scores become out of scale.

To get insights about the relationship between the average episodic returns of the final agents, and the predictive error of the underlying models, we propose the experiment in Figure 5. For all prediction horizons $h \in \{1, 5, 10, 25, 50\}$, we can see that in the medium dataset, the $R2(h)$ scores of the different models are positively correlated with their corresponding return. However, in the noisy dataset the relationship is not obvious. The interpretation of this empirical finding is not trivial as many factors are involved. However, a potential explanation is that related to the noise level of the environments. Indeed, models trained on noisy datasets may overfit the noise leading to worse generalizability, and consequently, worse agents in terms of the episodic return.

## 5.4 DYNAMIC EVALUATION: ITERATED BATCH RL

Finally, we run the full iterated batch MBRL (iterating between model learning, agent learning, and data collection) with the best multi-timestep models and compare the results with the vanilla one-step models in Figure 6. In this setting, the improvement brought by the multi-timestep models is not significant as the episodic return show a large variance among all the models.

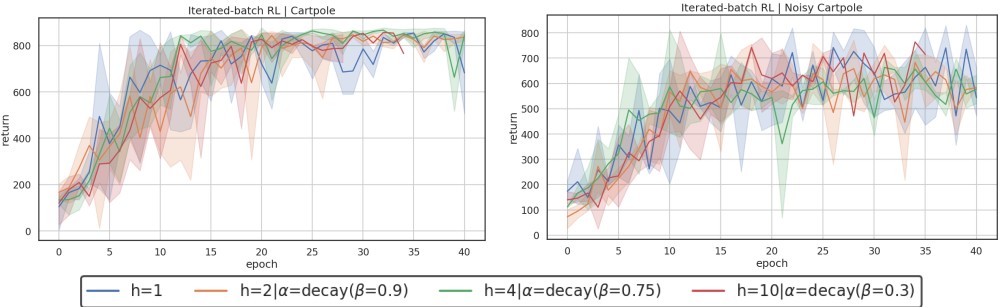

Figure 6: The iterated-batch RL results for the different multi-timestep models in comparison with the standard one-step model. We report the mean $\pm$ standard deviation across 3 seeds. The *epoch* in the x-axis denotes one iteration of the MBRL loop: model training on the dataset collected so far + Agent learning on model rollouts + evaluating the agent in the real system and updating the dataset.

## 6 CONCLUSION

In this paper, we ask how to adjust the training objective of one-step predictive models to be good at generating long rollouts during planning in model-based reinforcement learning. We build on the observation that fixed-horizon models achieve a lower predictive error down-the-horizon, and adapt the idea in the form of a novel training objective for one-step models. Our idea, which materialises in a weighted sum of the loss functions at future prediction horizons, led to models that, when evaluated against noiseless and noisy datasets from the Cartpole benchmark, show a large improvement of the R2 score compared to the vanilla one-step model. This improvement translates into better agents in pure batch (offline) RL. While the method did not improve the performance in the iterated batch setup, it did not lower the performance either. We will analyze in future work what confounding factors did not make performances improve.

One of the main research directions that follow-up from this work is the exploitation of gradient approximation schemes in the case of deterministic models. Indeed, as demonstrated in appendix C, the analytical form of our proposed loss function simplifies nicely as a linear function of the standard one-step loss. Furthermore, we plan to continue evaluating the resulting multi-timestep models in different settings/environments, and test their integration with state-of-the-art MBRL algorithms that rely on single-step dynamics models.

### LIMITATIONS

There are two major limitations of this study that could be addressed in future research. First, the study focused solely on the Cartpole benchmark, a relatively low-dimensional environment that reduces the modelling complexity that may be encountered in other high-dimensional systems. Nevertheless, to improve the generalisability of our results, we used different offline datasets and considered a more challenging Cartpole variant with additive Gaussian noise. Second, since our proposed objective can be plugged into any single-step model-based algorithm, we believe it would be interesting to test its integration with some of the state-of-the-art MBRL algorithms in different setups/applications. This is particularly important as our experiments in the offline and iterated-batch settings did not show a statistically significant improvement. Nevertheless, in this work we showcased the importance of multi-timestep models for the dynamics modeling task using different challenging setups, suggesting that they may well find RL applications, where the dynamics modeling is crucial for the final task.

## REPRODUCIBILITY STATEMENT

In order to ensure reproducibility we will release the code at `<URL hidden for review>`, once the paper has been accepted. The implementation details and hyperparameters are listed in Appendix B.

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
