## A    THE CARTPOLE REINFORCEMENT LEARNING BENCHMARK

### A.1    THE ENVIRONMENT

The Cartpole swing-up environment is characterized by a five-dimensional state space and a single-dimensional action space. The action in this environment consists of applying a horizontal force to the base of the cart, with the force ranging from -1 to 1. The task horizon is set to 1000. The reward function is designed to encourage the system to remain upright and centered, with small action and velocity values. The observables for this environment include the x-coordinate, the cosine and sine of the pole angle, the x velocity, and the pole's angular velocity. These parameters collectively define the dynamics and objectives of the Cartpole swing-up task.

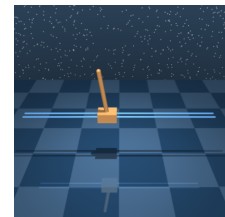

Figure 7: The cartpole environment.

### A.2    THE OFFLINE CARTPOLE DATASETS

In this section, we introduce the different datasets that are used to evaluate the multi-timestep models on the *Cartpole (swingup)* environment. These datasets are collected using some *behaviour policies* that are considered unknown to the agent (the combination of the world model and the actor that is using it for planning). All the datasets consist of $50$ episodes ($50k$ steps) split randomly as following: $36$ episodes for training, $4$ episodes for validation, and the remaining $10$ episodes are used for testing.

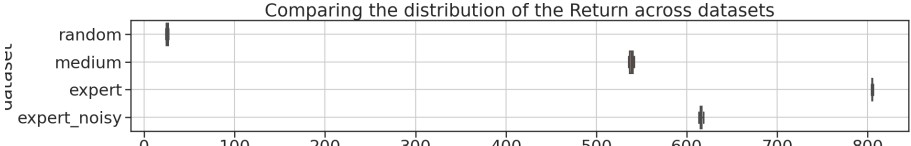

Figure 8: A comparison of the distribution of returns across the considered datasets.

We dispose of four datasets:

- **random.** This dataset is collected using a random policy (a policy that samples actions uniformly from the action space).

- **medium.** This is an intermediate between the two. It is collected by an unstable SAC agent that sometimes reaches near-optimal behaviour and sometimes doesn't manage to.

- **expert.** This is the full learning trace of an *expert*-policy (a Soft Actor-Critic -SAC- that is trained on an autoregressive mixture density network until convergence).

- **expert_noisy.** This is the full learning trace of the **expert** policy in the noisy Cartpole environment.

To understand the differences between these datasets we propose to analyze the variance of each state dimension of the Cartpole observables. We can see from Figure 9 that the **expert** dataset has very little variance as it converges quickly (around episode 8) and all the remaining episodes are almost identical. Similarly, the **random** dataset has little variance as it keeps exploring the same region around the initial state. The **medium** dataset, and naturally the **expert_noisy** dataset are the most diverse among the four, and we believe it would be insightful to evaluate the multi-timestep models against these different challenges, both in terms of the predictive error, and the final return of the underlying agents.

## B    IMPLEMENTATION DETAILS

For all the models, we use a neural network composed of a common number of hidden layers and two output heads (with *Tanh* activation functions) for the mean and standard deviation of the learned probabilistic dynamics (The standard deviation is fixed when we want to use the MSE loss). We use batch normalization (**?**), Dropout layers (**?**) ($p = 10\%$), and set the learning rate of the Adam

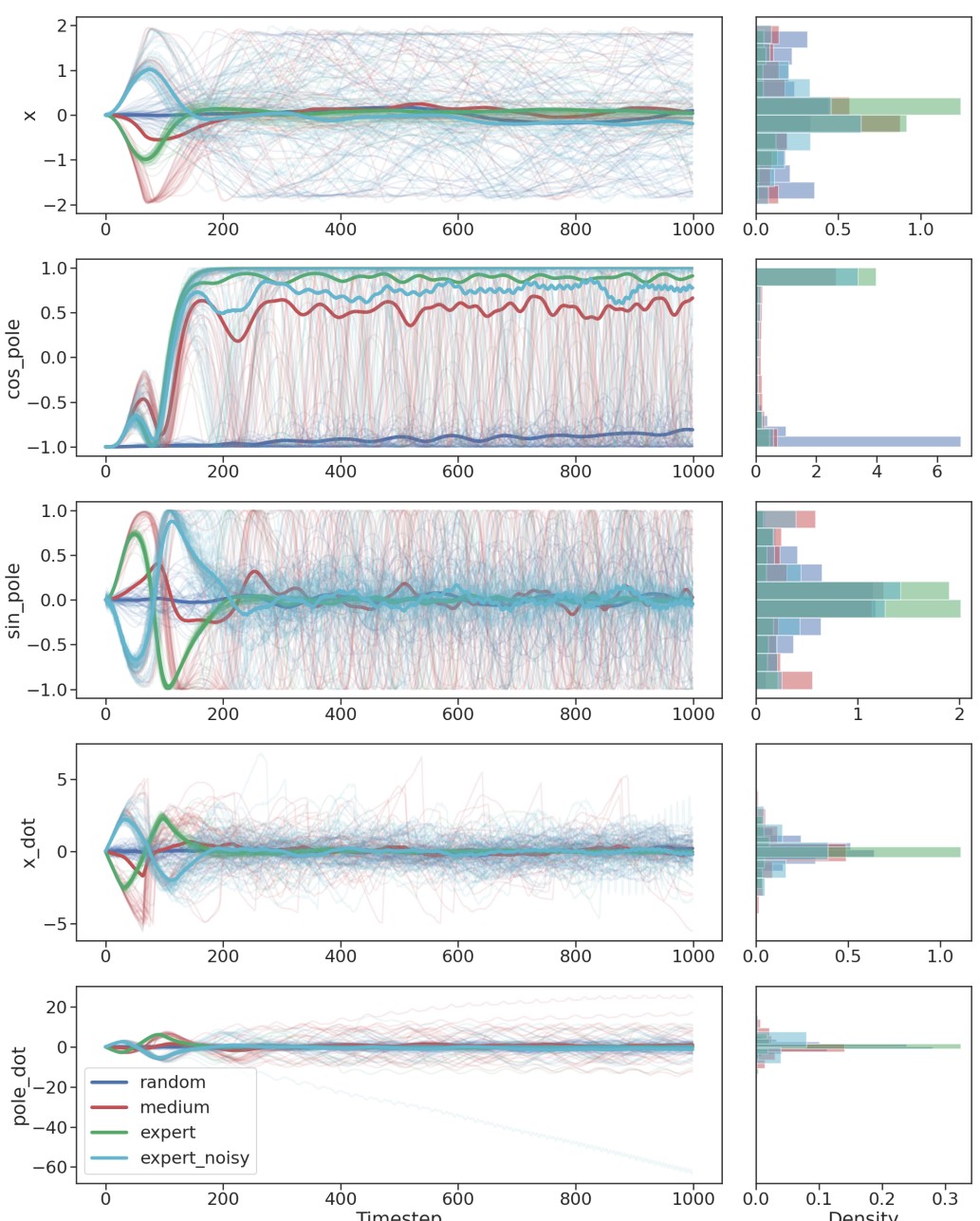

Figure 9: A comparison of the distribution of the state observables across the considered datasets.

optimizer (**?**) to $0.001$, the batch size to $64$, the number of common layers to $2$, and the number of hidden units to $256$ based on a hyperparameter search executed using the RAMP framework (**?**). The evaluation metric of the hyperparameter optimization is the aggregated validation R2 score across all the offline datasets.

For the offline and iterated-batch RL experiments, we use SAC agents from the open-source library StableBaselines3 (**?**) while keeping its default hyperparameters. In the offline setting, we train the SAC agents for $500k$ steps on a fixed model, and evaluate them by rolling-out an episode in the real environment. In the iterated-batch setting, the episodes generated from the evaluation of SAC are added to the current data buffer and used to retrain the model from scratch. For both these

setups, we modify the dynamics models to predict the state difference rather than the raw next state: $s_{t+1} \leftsquigarrow \hat{p}(s_t, a_t) + s_t$. This simple technique has been proved to improve the results of MBRL algorithms (Chua et al., 2018).

## C  GRADIENT COMPUTATION

In this section we want to compute the analytic derivative of the generalized loss: $L_{\alpha}(s_{t+1:t+h+1}, \hat{p}^{1:h}(s_t, a.)) = \sum_{j=1}^{h} \alpha_j L(s_{t+j}, \hat{p}^j(s_t, a_{t:t+j}))$ with respect to a model parameter $\theta$. For simplicity we denote the j-step loss $L^{(j)} = L(s_{t+j}, \hat{p}^j(s_t, a_{t:t+j}))$ and omit actions from the j-step prediction function: $\hat{p}^j(s_t)$. Under the assumption of a single-dimensional state and MSE loss, we derive the formula of $\frac{d}{d\theta} L^{(j)}$ for $j \in \{1, \ldots, h\}$:

$$
\begin{aligned}
\frac{d}{d\theta}[L^{(j)}] &= \frac{d}{d\theta}[(\hat{p}_\theta^j(s_t) - s_{t+j})^2] \\
&= 2\frac{d}{d\theta}[\hat{p}_\theta^j(s_t)](\hat{p}_\theta^j(s_t) - s_{t+j}) \\
&= 2\frac{d}{d\theta}[\hat{p}_\theta(\hat{p}_\theta^{j-1}(s_t))](\hat{p}_\theta^j(s_t) - s_{t+j}) \\
&= 2\frac{d}{d\theta}[\hat{p}_\theta^{j-1}(s_t)]\frac{d}{d\theta}[\hat{p}_\theta](\hat{p}_\theta^{j-1}(s_t))(\hat{p}_\theta^j(s_t) - s_{t+j}) \\
&= 2\left(\prod_{i=0}^{j-1}\frac{d}{d\theta}[\hat{p}_\theta](\hat{p}_\theta^i(s_t))\right)(\hat{p}_\theta^j(s_t) - s_{t+j}) \quad \text{(by recursion)}
\end{aligned}
\tag{3}
$$

From this we can compute the gradient of the loss $\frac{d}{d\theta} L_{\alpha}$:

$$
\begin{aligned}
\frac{d}{d\theta}\mathcal{L} &= \sum_{j=1}^{n} \alpha_j \frac{d}{d\theta} L^{(j)} \\
&= 2\sum_{j=1}^{n} \alpha_j \left(\prod_{i=0}^{j-1}\frac{d}{d\theta}[\hat{p}_\theta](\hat{p}_\theta^i(s_t))\right)(\hat{p}_\theta^j(s_t) - s_{t+j})
\end{aligned}
\tag{4}
$$

We start by observing that the terms featuring in the product are cumulative as we go further in the horizon. We thus compute the ratio between two consecutive loss terms $L^{(j-1)}$ and $\mathcal{L}^{(j)}$:

$$
\begin{aligned}
\frac{\frac{d}{d\theta}L^{(j)}}{\frac{d}{d\theta}L^{(j-1)}} &= \frac{2\left(\prod_{i=0}^{j-1}\frac{d}{d\theta}[\hat{p}_\theta](\hat{p}_\theta^i(s_t))\right)(\hat{p}_\theta^j(s_t) - s_{t+j})}{2\left(\prod_{i=0}^{j-2}\frac{d}{d\theta}[\hat{p}_\theta](\hat{p}_\theta^i(s_t))\right)(\hat{p}_\theta^{j-1}(s_t) - s_{t+j-1})} \\
&= \frac{d}{d\theta}[\hat{p}_\theta](\hat{p}_\theta^{j-1}(s_t))\frac{(\hat{p}_\theta^j(s_t) - s_{t+j})}{(\hat{p}_\theta^{j-1}(s_t) - s_{t+j-1})}
\end{aligned}
\tag{5}
$$

Given equation 5, we can write a formula for $\frac{d}{d\theta} L_{\alpha}$ that only depends on the prediction one-step ahead and its gradient. We denote the error ratio between horizons $k$ and $l$: $Err_{(k,l)} = \frac{(\hat{p}_\theta^l(s_t) - s_{t+l})}{(\hat{p}_\theta^k(s_t) - s_{t+k})}$; And the gradient of the prediction function evaluated at the $k$-th horizon: $G_k = \frac{d}{d\theta}[\hat{p}_\theta](\hat{p}_\theta^k(s_t))$. We use the convention that $G_0 = 1$ and $Err_{(0,1)} = 1$.

Let's refactor the expression of the gradient $\frac{d}{d\theta} L_{\alpha}$ in equation 4 using the ratio from equation 5:

$$\frac{d}{d\theta}L_{\boldsymbol{\alpha}} = \sum_{j=1}^{n} \alpha_j \frac{d}{d\theta}L^{(j)}$$

$$= \alpha_1 \frac{d}{d\theta}L^{(1)} + \sum_{j=2}^{h} \alpha_j \frac{d}{d\theta}[\hat{p}_\theta]\left(\hat{p}_\theta^{j-1}(s_t)\right)\frac{(\hat{p}_\theta^j(s_t) - s_{t+j})}{(\hat{p}_\theta^{j-1}(s_t) - s_{t+j-1})}L^{(j-1)}$$

$$= \alpha_1 \frac{d}{d\theta}L^{(1)} + \sum_{j=2}^{h} \alpha_j G_{j-1} Err_{(j-1,j)} L^{(j-1)}$$

$$= \alpha_1 \frac{d}{d\theta}L^{(1)} + \sum_{j=1}^{h-1} \alpha_{j+1} G_j Err_{(j,j+1)} L^{(j)}$$

$$= \alpha_1 \frac{d}{d\theta}L^{(1)} + \alpha^2 G_1 Err_{(1,2)}\frac{d}{d\theta}L^{(1)} + \sum_{j=2}^{n} \alpha_j G_{j-1} Err_{(j-1,j)} L^{(j-1)}$$

$$= \sum_{j=1}^{h} \alpha_j \left(\prod_{i=0}^{j-1} G_i\right)\left(\prod_{i=0}^{j-1} Err_{(i,i+1)}\right)\frac{d}{d\theta}L^{(1)} \quad \text{(by recursion)}$$

$$= \sum_{j=1}^{h} \alpha_j \left(\prod_{i=0}^{j-1} G_i\right)\left(Err_{(1,h)}\right)\frac{d}{d\theta}L^{(1)} \quad \text{(Elements simplify by definition of } Err_{(i,i+1)})$$

$$(6)$$

This formulation expresses the generalized loss derivative as a linear function of the derivative of the loss one-step ahead. The latter is relatively cheap and can be computed with one forward (and backward) pass through the model. Consequently, we can think of approximation schemes to reduce the computational burden $O(n \times Cost(L^{(1)}))$ that comes with computing the full derivative: $\frac{d}{d\theta}L_{\boldsymbol{\alpha}}$. We leave the exploration of this idea to a future follow-up work.