# OpenReview forum: "Multi-timestep models for Model-based Reinforcement Learning"
_ICLR.cc/2024/Conference — Submitted to ICLR 2024_

### Official Review · Reviewer_pXQz · 2023-10-29

**Soundness:** 1 poor
**Presentation:** 3 good
**Contribution:** 2 fair
**Rating:** 3
**Confidence:** 4

**Summary:**

The paper analyzes multistep prediction models for model-based reinforcement learning.
Training models for multiple steps can yield better performance. However, as every timestep step prediction is required for planning,  different loss compositions for training single-step models with multistep losses are studied.
The paper reports that an exponential weighting of loss terms, where longer time horizons have a lower weight, performs better, both in short as well as long-term predictions. The empirical evaluation is based on the cart-pendulum system.

**Strengths:**

- topic is important
- analysis and proposed loss weightings make sense
- analysis and methodology in principle good

**Weaknesses:**

- literally, no conclusion can be drawn from a single environment (with two levels of noise)
- the paper is empirical, so in order to make a robust statement the method needs to be evaluated on a large range of tasks
- the exponential weighting requires another hyperparameter that needs to be tuned to the horizon (and likely the task)
- presentation of results can be improved

**Questions:**

- Sec 4.2 + Fig 1: it is hard to believe that the model with h=10 makes no difference in the random case, but h=50 suddenly makes a big difference.
    - The inset does not really help, as all lines coincide
- Sec 4.3 + Fig 2: I am confused. Here $h=2$ ($\alpha=0$)$ improves performance whereas in Fig 1 it did not?!
    - Also, I would suggest plotting these results with respect to h=1. So h=1 is a flat line at 0. It would be potentially easier to see differences.
    - I think the long-term behavior (h>50) might not be so helpful, as such low $R^2$ values render the models useless anyway

- Sec 5.2: I would expect that when I need a model for h-step horizon planning, I would train it with an h-step loss (and then also only evaluate the prediction power up to h-steps. In the plots, there is mostly no visible difference after those steps. But how important is a difference in $R^2$ close to 1? Your Fig 5 suggests small differences can be actually quite important when planning.
- Fig 4: for h=4 and h=10, the pink lines seems to be missing
- Also for the non-noisy case, the learned weighting seems very competitive, which is really interesting.

- Sec 5.3:
I see where little robust results here. for h=10 it seems to be mostly worse and with high variance.
Fig 5: Do I understand correctly that e.g. the red markers are all the same model trained with n=10 loss but used only for planning with h=1, 5, 10... ?

Comments:
- I suggest in a future submission to run the same method on a variety of tasks going beyond just low-dimensional tasks and the DM-control suite. Following up on the learned alphas might be very fruitful and general.
- testing the strength of the models with a short horizon model-based planning method might also be useful

---

> ### Author Response · Authors · 2023-11-22
>
> We thank Reviewer pXQz for their insightful comments. Here we address some of their questions, and leave the open ones for a future submission.
>
> Sec 4.3 + Fig 2: Here $\alpha = 0$ is not equivalent to $h=2$ in Fig 1. $\alpha = 0$ corresponds to optimizing a one-step model $\hat{s}_{t+1} \sim \hat{p}(s_t, a_t)$ with the loss two-steps ahead:
>
> $L\Big(s_{t+2},\hat{p}\big(\hat{p}(s_t, a_t), a_{t+1}\big) \Big)$, While $h=2$ is a direct two-steps model $\hat{s}_{t+2}$
>
> $\sim \hat{p}(s_t,a_t,a_{t+1})$.
>
> Fig 4: The pink lines (decay with $\beta=0.9$) for $h=4$ and $h=10$ are missing because the corresponding R2 score is outside the interval $(0,1)$.
>
> Sec 5.3: Indeed, in Fig 5, dots of the same color represent the same model (and therefor have the same return value). We show different manifestations of the same dot along the x-axis (R2 score) to showcase the relationship between the return and $R2(h)$ for different values of $h$.

---

### Official Review · Reviewer_huS2 · 2023-10-31

**Soundness:** 2 fair
**Presentation:** 2 fair
**Contribution:** 1 poor
**Rating:** 3
**Confidence:** 4

**Summary:**

This papers studies how the accuracy of a model used for model-based reinforcement learning depends on the prediction horizon it has been to trained to predict at. It proposes different heuristics to weight the prediction horizons into a unified loss, and studies on CartPole how the prediction performance is related to the expected return collected by the resulting agent after RL training.

**Strengths:**

Despite the severe limitations of the paper, the following are positive points about its perspective on model-based reinforcement learning:
- The problem of finding better loss functions for training models of the dynamics, considering the final use that the reinforcement learning algorithm will make of these models, is important and relevant to the community
- I find the approach based on weighting different prediction horizons in a different way to be promising.

**Weaknesses:**

Unfortunately, I believe that the current iteration of the paper lacks a sufficient level of rigor for the contribution to be ready for publication:
- Despite the paper says this is a limitation, I believe the fact that the study is only conducted using a single, extremely simple, environment reduces the scope of the paper to be so small to be irrelevant. I encourage the authors to consider a larger suite of benchmarks, (e.g., MuJoCo, Brax, Myriad, MinAtar, Atari), picking the one that best suites their computational constraints.
- The results on the performance of RL algorithms are not particularly meaningful or significative (e.g., looking at Figure 4 and 5, the performance of all approaches seems to be the same): this might actually be related to the fact that more complex or even just diverse environments might be required to have a better understanding of how dynamics accuracy is related to expected return collected by an agent.
- Empirical comparison with similar approaches that are mentioned in the related work is missing. For instance, it is unclear why one should use a weighted loss instead of learning a different model for each time horizon.

**Questions:**

- What are the results in other environments?
- What are the results of comparison with other baselines?

---

> ### Author Response · Authors · 2023-11-22
>
> We thank Reviewer huS2 for their insightful comments. The empirical comparison with similar methods from the literature (particularly the Data Augmentation idea used in several papers [1,2,3]) will be included in a future submission.
>
> [1] Abbeel et. al. (2005) Learning vehicular dynamics, with application to modeling helicopters
>
> [2] Venkatraman et. al. (2015) Improving multi-step prediction of learned time series models
>
> [3] Erik Talvitie. (2017) Self-Correcting Models for Model-Based Reinforcement Learning

---

### Official Review · Reviewer_yoeH · 2023-10-31

**Soundness:** 2 fair
**Presentation:** 2 fair
**Contribution:** 2 fair
**Rating:** 3
**Confidence:** 4

**Summary:**

This paper tackles the compounding of model prediction errors
in model-based reinforcement learning for long prediction horizons.
They approach this by rolling out the prediction from a model
for multiple time-steps, computing the error for each time-step,
weighting the losses, and backpropagating the loss through the
full rollout trajectory. They tested several weighting strategies,
such as a uniform weighting, exponentially decaying weighting,
weighting to normalize the loss magnitudes.

The work performed experiments on the cartpole task in both a batch RL
setting (where they learn the model from data, then optimize the
policy from this model), and in an iterated RL setting, where the
policy is deployed in the environment to gather more data, iterating
multiple times to improve the performance.  In the batch RL setting,
the results were not statistically significant, while in the iterated
RL setting there was no improvement over a regular 1-step model.

**Strengths:**

- The use of the R2 score for evaluation of the prediction accuracy
was nice, as it provides an interpretable metric.
- The literature review and discussion were OK.

**Weaknesses:**

- The experimental results are not substantial. The method is only
demonstrated on cart-pole, and there is no statistically significant
improvement. I am not convinced the method works effectively.

- In some of the datasets, the data is generated from a fixed policy,
and the one-step model is used to predict the state at time step $t+h$,
by sequentially applying the actions $a_t, a_{t+1}, a_{t+2}$, etc. that
were applied in the rollout. In practice, the actions may also be
correlated with the state transitions, so making predictions in such
a feedforward manner may lead to inaccurate predictions. It may be
better to consider both the cases when applying the actions in a
feedforward manner, as well as the case when the actions are computed
from the policy based on the predicted states, as these may be different.
A simple exmaple to see the difference is when the environment is noisy,
and a feedback controller is applied in the system. The feedback controller
would control the system to eliminate the noise, and keep the system stable.
But if the model is rolled out in a feedforward manner, the applied
actions are unrelated to the noise, and the prediction may not be stable.

**Questions:**

Much more substantial experiments (in many more tasks) would be
necessary to change my opinion. This would be a large change to the
current manuscript, which I think would be too large and would require
a resubmission. Also, I am not confident that the method will provide
an improvement.

How do the computational costs for the different horizon training
methods compare?

---

> ### Author Response · Authors · 2023-11-22
>
> We thank Reviewer yoeH for their insightful comments. The correlation between action sequences and the underlying state transitions, as highlighted by Reviewer yoeH, is indeed a significant issue. Our current evaluation method is an approximation that downplays the role of the controller in the dynamics function. We argue that this remains a fair comparison method as it is consistently applied across models. However, we agree that a thorough examination of the interaction between the controller and the visited states necessitates a reconsideration of the current evaluation methodology, beyond just the final return of the agent.
>
> Reviewer yoeH’s second point concerns the computational cost of the multi-step loss. This is a primary follow-up idea that we are considering, and we have presented the initial stages of it in Appendix C. Indeed, the analytical form of the multi-step loss gradient simplifies nicely, and we plan to explore approximation schemes that will reduce the current linear temporal complexity.

---

> > ### Comment · Reviewer_yoeH · 2023-11-23
> >
> > Thanks for the comment, best of luck in your future submission!

---

### Official Review · Reviewer_XNaE · 2023-11-01

**Soundness:** 2 fair
**Presentation:** 3 good
**Contribution:** 1 poor
**Rating:** 1
**Confidence:** 4

**Summary:**

The paper proposes to use a multi-step loss to train a dynamics model. Instead of training on the standard 1-step loss, the paper proposes to use the n-step loss that recursively backpropagates through each model update. The learned dynamics model is then evaluated via batch (aka offline) and iterated reinforcement learning on the cartpole model.

**Strengths:**

The paper is nicely written and very easy to follow.

**Weaknesses:**

The paper has two severe weaknesses, first the proposed approach has been evaluated multiple times and second the experimental evaluation is very limited.

1) Multi-step Losses:
If I understand the proposed multistep loss correctly, this multistep loss has been proposed and utilized very often. For example, see the references [1-4] and there are many more. I am quite certain that one could even go back to the older system identification literature that talks about the multi-step loss for linear system identification. [4] even provides an ablation study across multiple horizons and systems. Interestingly, the results of [4] paint a different picture that going longer in the horizon is not necessarily better in terms of reward and quite frequently even worse (see Figure 3). The evaluation metrics as well as the RL algorithm are different in [4], [4] uses a CEM-like planner using the learned model instead of an actor-critic, but the discrepancy requires further investigation (see point 2).

Could the authors please precisely elaborate on how their multi-step loss is different from the previous works, except for the different weighting approach?

2) Experimental Evaluation:
The paper only evaluates using the cartpole swing-up task which is quite limited and does not include discrete contact events. Therefore, the paper would need to evaluate on much more and more complex dynamical system. Again [4] shows quite nicely that the results/conclusions from a cartpole do not transfer to a more complex system. [4] shows that for the cart pole, the longer horizon is always better in terms of prediction error and obtained reward, the conclusion cannot be generalized to more complex systems, where longer horizons start to perform worse.

To make more claims about the multi-step loss the paper needs more evaluation of more complex systems that involve contacts.

[1] Hafner et. al. (2015). Learning latent dynamics for planning from pixels

[2] Abbeel et. al. (2005) Learning vehicular dynamics, with application to modeling helicopters

[3] Venkatraman et. al. (2015) Improving multi-step prediction of learned time series models.

[4] Lutter et. al. (2021). Learning Dynamics Models for Model Predictive Agents

**Questions:**

see above

**Post Rebuttal Comment:**

I thank the reviewers that they accepted the reviews and did not try to convince the reviewers that they are wrong in the rebuttal. As the rating of all reviewers is very coherent, I keep my score.

---

> ### Author Response · Authors · 2023-11-22
>
> We thank Reviewer XNaE for their insightful comments. We now delve into a discussion on the references cited by Reviewer XNaE and how they intersect with our contribution:
>
> [1] Hafner et. al. (2015). Learning latent dynamics for planning from pixels: This paper presents a derivation of a multi-step $\beta$-VAE bound on latent sequence models trained via variational inference (termed latent overshooting). While the underlying intuition is similar, the context in which we apply the multi-step loss differs, focusing on simple feedforward dynamics models trained via MSE minimization. Moreover, the authors discovered that the latent overshooting variant of their algorithm didn’t yield any improvements and was not adopted in their final agent. This could potentially be due to the fixed value of all $\beta_{>1}$​, which overlooks the exponential growth of the error, a problem we aim to address in our study.
>
> [2] Abbeel et. al. (2005) Learning vehicular dynamics, with application to modeling helicopters: This paper indeed proposes a similar loss function termed “lagged criterion” (equation 4). However, to simplify the non-convex optimization to a linear least squares problem, the authors relax the dependency of intermediate predictions $\hat{s}_{t+h}$ on the model parameters $(A,B)$. This results in a data augmentation-like technique that instructs the model to recover from its own errors without propagating the gradients through the multi-horizon applications of the model.
>
> [3] Venkatraman et. al. (2015) Improving multi-step prediction of learned time series models: Although this paper begins with a theorem (Theorem 1) that illustrates the exponential growth of the multi-step loss, the algorithm idea “Data-As-Demonstrator” is essentially a data-augmentation procedure that “synthetically generates correction examples for the learner to use”. This is akin to [2] and does not involve any gradient-based optimization of the multi-step loss as we propose.
>
> [4] Lutter et. al. (2021). Learning Dynamics Models for Model Predictive Agents: This paper is the closest to our concept as it back-propagates the gradients through the multi-step loss, and we thank Reviewer XNaE for highlighting this. However, as Reviewer XNaE pointed out, the differences in the model architecture, the evaluation metric, and the actor could potentially be confounding factors in our respective analyses. Furthermore, the question of the optimal horizon is strongly tied to the weighting of the intermediate MSE terms as this corrects the differences in scale. This question is not considered in [4], and we aim to demonstrate that it’s a crucial component in maximizing the benefits of the multi-step loss component.

---

### Author Response · Authors · 2023-11-22
**General comment**

We thank the reviewers for their insightful feedback on our paper. We recognize the reviewers’ concerns regarding the limited experimental scope that supports our idea. We intend to incorporate the various points raised by the reviewers in a future submission, including comparisons with existing literature and different tasks/environments.

---

### Meta-Review · Area_Chair_XCHo · 2023-12-09

**Metareview:**

This paper is about training multi-step dynamics model for model-based RL. The paper formulates weighted loss functions for dynamics at each timestep, examining different variations of the proposed weights (eg learned, or fixed uniform). The weighted loss is optimized through gradient-based optimization to train the dynamics. The performance of the learned dynamics model is evaluated according to its supervised learning performance; the performance of MBRL using only transition data from the model; and the performance of MBRL using both environment transitions and model transitions. The method is evaluated on the cartpole environment.

While the paper deals with a very interesting topic, the main issues with the current version fo the work is that (a) evaluation is only limited to the cartpole environment and (b) its novelty is questioned by reviewers, given the methods proposed in [3,4].

The authors' responses to reviewers did not fully address these issues. This would be a considerably stronger paper if (a) was addressed.


[3] Venkatraman et. al. (2015) Improving multi-step prediction of learned time series models.
[4] Lutter et. al. (2021). Learning Dynamics Models for Model Predictive Agents

**Justification For Why Not Higher Score:**

Because I don't think the paper is ready for publication.

**Justification For Why Not Lower Score:**

N/A.

---

### Decision · Program_Chairs · 2024-01-16

Reject